# Study on the Status of Scientific Fitness Literacy of Rural Left-Behind Minors and the Influence of Family Environment

**DOI:** 10.3390/ijerph20010249

**Published:** 2022-12-23

**Authors:** Xiang Pan, Yibo Gao, Lupei Jiang, Pengyu Deng, Jin He, Tian Xia, Aoyu Zhang, Yanfeng Zhang

**Affiliations:** 1China Institute of Sport Science, Beijing 100061, China; 2Graduate School of Health and Sports Science, Juntendo University, Inzai 270-1695, Japan; 3Sports Coaching College, Beijing Sports University, Beijing 100084, China

**Keywords:** left-behind minors, rural, scientific fitness literacy

## Abstract

There are a large number of 6.97 million left-behind children in China at the end of August 2018. Left-behind children exhibit many psychologically unhealthy behavioural disorders. This study aimed to compare the differences in scientific fitness literacy (SFL) between rural left-behind and general minors, and to analyze the factors affecting the SFL of left-behind minors in the family environment. A random stratified sampling was conducted among minors aged 3–18 in rural areas of China, and 2239 valid samples were obtained. The questionnaire is based on the SFL part and the family environment part of the China National Fitness Activity Status Survey, except for the SFL part of the children’s questionnaire does not contain the attitude dimension, the questionnaires of several other age groups contain three dimensions: attitude, skills, and habits. The SFL of general children and adolescents was higher than that of left-behind children and adolescents, and in the sub-dimensions, children’s attitudes and adolescents’ skills and habits differed greatly, with the general minors having better performance. The SFL showed a significant increasing trend with age in the general minors but the difference between children and adolescents in the left-behind minors was not significant. The scores of habituation in left-behind minors increased with age group, and the scores of the general minors group did not vary much between age groups. In the multiple regression analysis for the left-behind minors, it was found that left-behind young children were less influenced by family environment on SFL score, and children’s SFL was more influenced by family environment. A separate analysis of the 7–9 and 10–12-year-old groups revealed that factors such as parental support and encouragement influenced SFL with different weights, and the importance of parents rotated, while the number of sporting goods in the family was important in all four age models. For the left-behind minors with a large amount in China, the role of the family environment in their SFL cannot be ignored, and the promotion of this literacy should be carried out in a targeted manner.

## 1. Introduction

Scientific fitness literacy (SFL) is a concept that more comprehensively reflects the causes, processes, and outcomes of people’s participation in physical activities and exercise. It refers to the ability of people to reach their embodied potential scientifically according to their own specialty, actively interact with their environment, and build a virtuous cycle from motivation, knowledge and understanding, attitudes, abilities and skills, to behaviors and habits, and eventually, they can be able to consciously participate in activities related to sports at a certain level throughout their lives [1]. SFL is more precisely focused on exercise and fitness rather than overall physical activity levels and physical literacy.

As an important place for the development of the individual both physically and mentally, family plays a pivotal role in all aspects of children’s development. Parents are not only the most important resource for children’s psychological development but also the starting point and cornerstone for the healthy psychological development of their children. The influence of family environment on SFL in 3–9-year-old children has been analyzed in depth [2], in that behavioral factors such as parental example, involvement, encouragement and persuasion, instrumental support and restriction, and the level of parental education all have an impact on children’s SFL score. Moreover, parental education provides more opportunities for teenagers to engage in healthy behaviors [3], and also parental social status [4] and sports participation [5] have some effects on children’s sports participation. However, in China, there are many minors who are in foster care with relatives or grandparents. According to the China Child Welfare and Protection Policy Report in 2019 [6], there are a large number of 6.97 million left-behind children in China at the end of August 2018. Especially in recent years, as urbanization advances, rural laborers go out of their hometowns to work but are restricted by objective conditions such as job instability and educational resources, so some choose to leave their minor children in their hometowns under the supervision and care of others, resulting in the emergence of a large number of rural left-behind minors. Due to the lack of parental care and family education, left-behind children exhibit many psychologically unhealthy behavioral disorders. In addition, given the increasing rate of dual-worker families, Chinese families rely heavily on (maternal) grandparents or relatives and friends to care for children, and as a result, many minors have less parental involvement in their lives [7]. Most left-behind children are raised and educated by intergenerational parents, and the quality and level of their family education directly affect the healthy growth of left-behind children. In recent years, the problem of intergenerational family education for left-behind children has emerged frequently, reflecting that the phenomenon of intergenerational family education deserves our attention and research [8]. The SFL for rural left-behind children should also be a concern nowadays. It is unknown whether the SFL of left-behind minors is different from that of general minors, and whether the influence of family environment on minors’ SFL is not influenced by their left-behind status or its mode of action is changed. Therefore, this study compares the differences in SFL between left-behind minors and their general peers with the help of a survey on the current situation of SFL among rural minors in China, and further analyzes in depth the effects of family environment on rural left-behind minors at different ages, in order to provide a theoretical basis and new ideas for the subsequently targeted formulation of decisions and guidelines to improve SFL among rural left-behind minors in China from the perspective of family environment.

## 2. Materials and Methods

### 2.1. Sampling and Survey

A national county (county-level city/district/banner) sampling frame was constructed based on the data from the sixth population census, and proportional probability sampling (PPS) to the population size was used. Ten to twenty villages were randomly selected from the county (county-level city/district/banner) sampling frame of each province (autonomous region and municipality) on the basis of differences in socioeconomic development levels while taking into account the geographical location of the province, respectively, and each village was randomly selected from 3–6-year-old, 7–9-year-old, 10–12-year-old, and 13–16-year-old minors in each case. Based on the Provincial Institute of Sports Science, survey teams at all levels supervised the quality of survey work and data aggregation in each city and county. Furthermore, in this study we refer to the Opinions on Strengthening the Care and Protection of Children Left Behind in Rural Areas issued by the State Council [9] which defines two types of min ors as left behind: Firstly, those who cannot be supervised and cared for because of their parents both away for work; secondly, those where one parent is away for work and the other is incapable of supervising and caring for them; all the rest are ordinary.

The survey method of this study was a household survey, and for the convenience of registration of population information, the sampling frame of this survey was established by households, but the actual sampling was conducted by people, so there was a possibility that someone was sampled for the survey among the same sampled households.

### 2.2. Test Instrument

The questionnaire is based on the SFL section of the China National Fitness Activity Survey and the family environment section of the survey, in which we set two sets of questions for children with slightly different contents. The difference is that the 7–9-year-old section takes into account the weak vocabulary and comprehension ability of primary school students in the lower grades, so the questions are mainly filled in by children and parents, and complex questions can be answered with the assistance of parents, while abstract questions are answered by parents on their behalf. Ages 10–12 are upper elementary school students and the questionnaire was filled out directly by themselves.

Except for the SFL section of the young children questionnaire, which does not contain the attitude dimension, the questionnaire for several age groups contains the three dimensions of attitude, skills, and habits, and the reliability of this questionnaire has been verified to reflect the level of SFL and its sub-dimensions of minors [2,10]. Attitude (emotional level, including value judgments, emotional likes or dislikes, and behavior tendencies), ability and skills (physical level, including motor and exercise skills, recognition, and self-protection skills), behavior and habits (behavior level, including the frequency, intensity, duration, and persistence of the behavior). The above contents are latent variables reflecting scientific fitness literacy, and each sub-factor corresponds to a scale. Family environment aspects according to the results of Sallis, etc. [11,12,13,14]. We divided the family environment into two aspects—basic information about family environment and family behavior [11]. Basic information about family environment includes three indicators: parents’ education level, parents’ income level, and whether parents are the primary caregivers. Family behavior comprises five indicators, which are modeling, involvement, encouragement and persuasion, instrumental support, and restriction. Some examples of questions in the family environment questionnaire are shown in Table 1.

### 2.3. Procedure

Full-time surveyors from the municipality’s sports management office were trained to familiarize themselves with the protocols of the study. Information on respondents at the monitoring sites was obtained from the local statistical office and they were randomly selected to meet the criteria. They were contacted by telephone and completed the information in a face-to-face questionnaire after obtaining their consent. A face-to-face interview lasting approximately one hour per participant was conducted in the community at the designated study site, after which the information was completed on the questionnaire. Quality control was also carried out during the survey, including follow-up phone calls and on-site recording. Quality control during the survey was carried out by callbacks and field notes. The survey was conducted from June to August 2020. In this study, names were replaced with Arabic numerals, the entire sample collected did not contain any personally identifiable information, and parents of minor participants were briefed on the procedures and objectives of the study before the beginning of the study, and written consent was obtained with their children. Ethical approval for this study was obtained from the China Institute of Sport Science (Beijing, China).

### 2.4. Analysis Method

In this study, the three comparisons between groups were calibrated by using the Bonferroni test, *p*’ = *p* × (1/3) to take the test level α = 0.05, i.e., *p*’ < 0.0167 as a significant criterion. We took the scientific fitness literacy as the dependent variable, and all influencing sub-factors of family environment (see Table 1) studied previously as independent variables to establish multiple linear regression models for each age group and used stepwise regression analysis to screen all influencing sub-factors of family environment to obtain the optimum model for left-behind minors in each age group. To analyze which type of family environment or influencing sub-factor plays the main role for left-behind minors in each age group.

## 3. Results

### 3.1. Descriptive Statistics

Table 2 shows the descriptive statistics of this study. In this survey, we delivered a total of 2400 questionnaires, and a total of 2239 were collected, with a recovery rate of 93.3%. The proportion of males and females in each age group was relatively balanced, and the proportion of general minors and left-behind minors groups differed significantly due to natural sampling.

### 3.2. Multi-Age Comparison of General and Left-Behind Minors’ Scientific Fitness Literacy and its Sub-Dimension Scores

As shown in Table 3, the SFL scores of the left-behind minors showed higher scores with age, but the SFL scores of general adolescents were slightly higher than those of children with no significant difference. Both general and left-behind minors showed higher skill dimension scores for children than for young children but also significantly higher than that of adolescents. This phenomenon also appears in the scores of left-behind minors’ habit dimension, but there is no significant age difference in the scores of ordinary minors’ habit dimension.

There were no significant differences between the SFL scores and their sub-dimension scores of general and left-behind young children. The SFL scores of general children were significantly higher than left-behind children for all sub-dimensions except in the habit sub-dimension, especially for the attitude dimension while the SFL scores of general adolescents were significantly higher than left-behind adolescents for the sub-dimensions of skill and habit, but not for the attitude dimension. There were no significant differences in the attitude dimension. It can be seen that parental care was not significantly related to SFL for young children, but more closely related to SFL for children and adolescents, where general children tended to have better attitudes and skills and dimensions, and general adolescents had higher skills and habits.

### 3.3. Multivariate Regression Model of the Impact of Family Environment on the Scientific Fitness Literacy of Left-Behind Minors in Multiple Age Groups

We modeled SFL based on questionnaire data about the family environment for left-behind minors in each age group, respectively. As shown in Table 4, very few factors entered the multiple regression model for SFL of left-behind young children aged 3–6 years with low explanatory power (adjusted R^2^ = 0.114), only the number of sports facilities and equipment owned by the household and time spent with parents. Both factors were positive for SFL in left-behind young children, and the β difference between them was not significant.

The number of sports facilities and equipment owned by the household had the largest β value in the 7–9-year-old model (β = 0.530), and this factor also appeared in the 10–12-year-old group model (β = 0.321). Parents’ encouragement and support entered the 7–9-year-old model and the 10–12-year-old model, but it had the highest weight in the 10–12-year-old model (β = 0.439) and the second highest weight in the 7–9-year-old model (β = 0.391). It is rather peculiar that the number of electronics owned by the household plays a positive role in the 7–9-year-old model (T = 2.432, *p* < 0.05) while the number of sports facilities and equipment does not enter the model. In the 10–12-year-old model, the number of sports facilities and equipment plays a positive role (T = 4.582, *p* < 0.01). Mothers’ education only entered the 7–9-year-old model while fathers’ education only entered the 10–12-year-old model, which can be seen as a result of quasi-switching the importance of parental roles.

Factors regarding parental attitudes and encouragement still existed in the adolescent SFL model (β = 0.321), but the importance of the family’s economic strength was higher (number of sports facilities and equipment owned by the household, β = 0.439; parents’ willingness to pay for their children to take sports-related classes, β = 0.168).

## 4. Discussion

In the current study, we surveyed rural minors in 31 provinces (municipalities and corps) in China and compared the SFL and its sub-dimension scores of general and left-behind minors by age, followed by multiple regression models to analyze the influencing factors for each age group of left-behind minors. We found that the SFL of general children and adolescents was higher than that of left-behind children and adolescents, and in the sub-dimensions children’s attitudes and adolescents’ skills and habits differed greatly, with the general minors having better performance. The SFL showed a significant increasing trend with age in the general minors but the difference between children and adolescents in the left-behind minors was not significant. The scores of skill dimensions were higher in children than that in young children, but the scores of left-behind adolescents were extremely low. The scores of habituation in left-behind minors increased with age, and the scores of the general minors group did not change much between age groups, but the scores of general minors were higher than those of the left-behind minors within the same age group. In the multiple regression analysis for the left-behind minors, it was found that left-behind young children were less influenced by family environment on SFL score, and left-behind children’s SFL was more influenced by family environment. A separate analysis of the 7–9 and 10–12-year-old groups revealed that factors such as parental support and encouragement influenced SFL with different weights, and the importance of parents rotated, and electronic items in the family appeared to be one of the positive factors of SFL in children aged 7–9 years, while the number of sporting goods in the family was important in all four age models.

### 4.1. Scientific Fitness Literacy Differs between Rural Ordianry Minors and those Left behind and the Difference Varies with Age

With age, SFL, habitual dimensions improve while skills and attitudes decline. In the national fitness activity surveys of 200,000 people researched by the National Center for Physical Fitness Monitoring in 2014 and 2020, it was shown that in China, not only the percentage of people who participate in physical activity at least once a week among minors increases with age, but conversely, this percentage gradually decreases with age among people over 19 years old, including both adults and elder people. The reason for the decrease in scientific fitness attitudes is intriguing and is related to the positive effect of parental physical literacy on children’s values of physical activity [15] plus the decrease in the explanation of SFL by the family environment we found. Except for young children, SFL scores were lower for children and adolescents in the left-behind minors than that in the general minors. Children were reflected in attitudes and adolescents in skills and habits. In the Chinese survey, it was found that rural left-behind children do not have a subjective understanding of participation in sports activities outside school [16], and their attitudes toward school-organized sports activities, physical education classes, and physical exercise effects were not positive [17], and some scholars concluded through ordered logistic regression analysis that the influencing factors affecting left-behind children’s participation in physical exercise found that the family factor in which parents accompanied their children for the highest ratio of physical exercise was found in the family factor, as the parents were away for a long time and did not accompany their children enough, and the intergenerational guardianship did not know enough about physical exercise, which caused the left-behind children to have low willingness to exercise and low awareness of physical exercise [18]. In contrast to the differentiation of attitudes, motor habits and skills show a certain lag, with the frequency of play in preschool and exercise in childhood significantly influencing subsequent motor habits and physical fitness levels [19], and not only that, but family influence also shows a certain lag, with parental well-being, family screen time and socioeconomic status in early childhood also predicting children’s motor habits in the 8–13-year-old stage. For the left-behind minors, family influences were not significant in early childhood, with no significant differences in SFL scores on the one hand, and on the other hand, the model interpretation was not high. The degree to which SFL is influenced by family environmental factors decreases with age during childhood and adolescence. The socialization of the individual may have contributed to this change, with the role of school and community taking over the influential position of the family environment [1].

### 4.2. Parental Roles Still Have an Impact on the SFL of Rural Left-Behind Minors

The role of parents begins to emerge more in the childhood stage. Although Paez, J., etc. concluded that parents’ physical activity levels are not directly related to children’s nutritional status or motor development [20], most studies have shown that parental involvement has a positive effect on children’s physical participation and attitudes, and that parental values and support for PA mediates the relationship between parents’ physical literacy and children’s values for PA, so interventions to improve parents’ physical literacy and parental support for PA in the home environment can be used to effectively improve children’s values for physical activity [15]. Parental encouragement and support for children’s sports, children’s sports participation, the amount of time children spend outdoors, community spaces for physical activity, and the number of sports facilities nearby home were all positively associated with children’s physical activity from grade 5–7 [21], which coincides with our findings. Parental education had a positive effect on SFL at the childhood stage in the left-behind minors, with higher education levels having a greater awareness of physical activity, a higher percentage of those who had participated and regularly participated in physical activity, a higher annual per capita amount spent on physical activity, and a lower percentage of those who took small intensities for physical activity [22], also being major positive influences for children’s health-related behaviors [23], corresponding to the fact that their studies have shown a quantifiable relationship between the number of steps/day of parents and children. With mothers’ steps/day increasing by 263–439 for each additional 1000 steps for their sons and 195–219 additional steps/day for their daughters [24], and not only that, both mothers’ and fathers’ role models positively influenced children’s physical activity in leisure time, parental involvement in sport and exercise is significantly and positively correlated with the number of hours of leisure time participation in sport for boys and girls respectively [25]. Whereas the education level of adults was positively associated with their frequency of sports [26], higher-educated parents correspondingly had an indirect positive contribution to their children’s frequency of sports or with, therefore.

### 4.3. Compared to Total Income, Sports Expenditure Explains the SFL of Rural Retained Minors Better

Unexpectedly, annual household income was entered into the model of family factors affecting the SFL of left-behind minors at all ages. In developed countries, preschoolers from low-income settings performed worse in gross motor skills compared to their peers from high-income settings [27], and due to less access to opportunities to develop motor skills [28], they had less access to appropriate education and practice and had a greater probability of developmental lag [29]. However, the correlation between SFL and household income among minors of all ages in our study was not demonstrated, which may be related to the small share of sports consumption in the overall consumption of rural residents, as shown in *2020 China Residents Sports Consumption Development Report* released by the General Administration of Sport of China, which showed that sports consumption in rural China accounted for 6.54% of rural per capita consumption in 2020, a decrease from 2007. The SFL of left-behind minors is positively influenced by the amount of sports equipment in the family environment but not by household income, which also reflects that perhaps sports consumption is one of the factors in income level that affects children’s SFL and basic sports participation such as sports skills and frequency.

### 4.4. Highlights and Shortcomings

This study is based on a nationwide rural cross-sectional survey, with a high scientific sample size and a strong national representativeness, but the sample size of this survey is not large because it is a survey of national governmental public interest research in advance so that it is limited by the lack of human and material resources. This study focuses on minors whose first caregivers are not parents, i.e., “left-behind minors”, which is a special group drawing high social attention in China. The present study fills the gap in this area by looking at the influence of family environment. In addition, not only in rural areas, but also in towns, there are a certain number of left-behind minors. To ensure the relevance of the study, this study did not investigate and study this group, and the follow-up study can enlarge the sample of left-behind minors not only in rural areas to verify and compare the difference between urban and rural areas, in order to enrich the theoretical basis and practice of interventions on SFL of left-behind minors in their family environment. The study is expected to provide a theoretical basis and a practical way to expand interventions for SFL among left-behind minors in their family environment.

## 5. Conclusions

There are large differences in SFL levels between rural left-behind minors aged 3–18 years old and those of general minors, both in comparison between different age groups and in different sub-dimensions. The family environment has little impact on the SFL of left-behind children in rural areas, and it begins to become prominent after childhood. Although the parental participation of rural left-behind minors is low in family environment, it still has a hugely positive effect, and the material foundation of the family continues to act positively on the whole stage of 3–18 years old, even if there is an age difference in importance. For the left-behind minors with a large amount in China, the role of the family environment in their SFL cannot be ignored, and the promotion of this literacy should be carried out in a targeted manner.

## Figures and Tables

**Table 1 ijerph-20-00249-t001:** Sub-dimensional factors of household environmental impact and questionnaire examples.

Family Environment	Sub-Dimensions	Examples of Questionnaires
Base information	Father’s education	Highest level of father’s education.
	Mother’s education	Highest level of mother’s education.
	Family income level	Gross household income for the most recent year.
	First caregiver	People who take care of children’s daily lives.
Family behavior	Modeling	Parental physical activity frequency.
		Things caregivers often do.
	Participation	Do your parents participate with you?
	Encouragement and Persuasion	Did your parents encourage or support you to participate in physical activity?
		Will your parents be willing to listen to your ideas and opinions?
	Instrumental support	What are the items you often touch at home (including sports equipment electronic devices)?
	Restrictions	What are your parents’ expectations and restrictions on your physical activity?

**Table 2 ijerph-20-00249-t002:** Basic information.

Age		3–6	7–9	10–12	13–18
		*n* (%)	*n* (%)	*n* (%)	*n* (%)
Gender	Male	253 (47)	242 (46)	284 (48)	301 (51)
	Female	287 (53)	280 (54)	306 (52)	286 (49)
Types of minors	General	416 (77)	468 (79)	437 (84)	496 (84)
	Left-behind	124 (23)	122 (21)	85 (16)	91 (16)

**Table 3 ijerph-20-00249-t003:** Scientific fitness literacy and sub-dimensional scores for minors.

	Caregiver Grouping	Young Children (*n* = 540)		Children (*n* = 1112)		Adolescents (*n* = 587)	
		Mean (SD)	*p*	Mean (SD)	*p*	Mean (SD)	*p*
Age	Left-behind	3.98 (0.96)	0.699	9.66 (1.64)	0.145	15.38 (1.77)	0.309
	General	4.01 (0.98)		9.48 (1.64)		15.19 (1.68)	
	Total	4.01 (0.97)		9.51 (1.64)		15.22 (1.70)	
SFL	Left-behind	33.37 (12.36)	0.732	52.02 (14.79) ^a^	0.012	55.34 (12.00) ^ab^	0.015
	General	33.82 (12.82)		54.83 (14.33) ^a^		58.84 (12.69) ^a^	
	Total	33.72 (12.71)		54.30 (14.45) ^a^		58.30 (12.64) ^ab^	
Attitude	Left-behind			68.71 (17.64)	0.003	68.46 (14.60)	0.318
	General			72.45 (16.05)		70.00 (13.39) ^b^	
	Total			71.75 (16.42)		69.76 (13.58)	
Ability and skills	Left-behind	20.03 (15.66)	0.604	25.08 (19.99) ^a^	0.037	19.20 (15.42) ^b^	0.007
	General	19.24 (14.43)		27.93 (21.55) ^a^		25.17 (20.09) ^ab^	
	Total	19.42 (14.71)		27.40 (21.29) ^a^		24.24 (19.55) ^ab^	
Behavior and habits	Left-behind	46.81 (15.71)	0.308	48.96 (18.24) ^a^	0.082	46.40 (18.49) ^ab^	0.001
	General	48.58 (17.32)		51.90 (18.17)		53.92 (19.08)	
	Total	48.18 (16.97)		51.35 (18.21) ^a^		52.75 (19.17) ^a^	

*p* is the test of significance of the difference between normal and left-behind minors. Comparisons with young children: a, *p*’ < 0.0167. Comparison with children: b, *p*’ < 0.0167.

**Table 4 ijerph-20-00249-t004:** Multiple regression models of the influence of family environment on the scientific fitness literacy of minors.

Factors that Enter the Model	F	Adjusted R^2^	T	β
Number of sports facilities and equipment owned by the household	8.947 **	0.114	2.727	0.242 **
Time with parents during holidays			2.292	0.203 *
Number of sports facilities and equipment owned by the household	26.691 **	0.550	6.883	0.530 **
Whether the parents encourage and support children’s participation in sports			5.176	0.391 **
Number of electronics owned by the household (mobile phones, computers, etc.)			−2.432	0.190 *
Mother’s education			2.189	0.162 *
Whether the parents encourage and support children’s participation in sports	23.900 **	0.429	6.117	0.439 **
Number of sports facilities and equipment owned by the household			4.582	0.321 **
Father’s education			2.349	0.168 **
Whether the parents are willing to listen to their children’s opinions.			2.089	0.156 *
Number of sports facilities and equipment owned by the household	20.460 **	0.393	4.017	0.439 **
Parents often engage in physical activity			3.091	0.321 **
Parents’ willingness to pay for their children to take sports-related classes			2.868	0.168 **

*: *p* < 0.01. **: *p* < 0.001.

## Data Availability

The data presented in this study are available on request from the corresponding author. The data are not publicly available due to privacy.

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
