# Peer review of "Study on the Status of Scientific Fitness Literacy of Rural Left-Behind Minors and the Influence of Family Environment"

_ijerph, 2022, doi:10.3390/ijerph20010249_

Round 1

Reviewer 1 Report

The authors start the introduction by providing information about the context of the study but then there is a gap in the ideas between the context and the aim of the study. In order to bridge this gap, please state clearly your research question in the introduction (between lines 69 and 70).

Although the authors refer to some literature to support and discuss findings, it could be enhanced (for instance, in the introduction) by adding a discussion / review about the concepts / variables under study. 

Although general information is provided about the development of the test instrument, some is missing. For instance, the authors mention the variables under study but they do not explain why these variables (and then the dimensions / categories used in the questionnaire). How were the answers measured? Likert scale? Other? How many questions? How did the authors manage to identify the general respondents and the left-behind? This is not completely clear.

Need proofreading.

Author Response

Response to Reviewer 1 Comments

The authors start the introduction by providing information about the context of the study but then there is a gap in the ideas between the context and the aim of the study. In order to bridge this gap, please state clearly your research question in the introduction (between lines 69 and 70).

Response 1: Sorry for the discrepancy between the presentation of the context and the research objectives due to the presentation problem, we have filled this gap (lines 70-79).

Although the authors refer to some literature to support and discuss findings, it could be enhanced (for instance, in the introduction) by adding a discussion / review about the concepts / variables under study. 

Response 2: We are sorry for your suggestion, scientific fitness literacy is the first variable in China's localization, we have analyzed and discussed it as much as possible, but there is not much literature to superimpose for the time being, after all, you can also see, we cite some of our previous research to analyze.

Although general information is provided about the development of the test instrument, some is missing. For instance, the authors mention the variables under study but they do not explain why these variables (and then the dimensions / categories used in the questionnaire). How were the answers measured? Likert scale? Other? How many questions? How did the authors manage to identify the general respondents and the left-behind? This is not completely clear.

Need proofreading.

Response 3: Regarding the measurement of scientific fitness literacy, we are sorry that we have missed some references before, which are now marked in line 114 of the text. We have also added an explanation on how to distinguish between ordinary and left-behind minors, in lines 92-97.

In the language section, we have corrected and polished the grammar of the manuscript. Thank you for your criticism.

Reviewer 2 Report

Thank you so much for giving me the opportunity to review this manuscript.

General Comments

The authors explored an important issue related to the scientific fitness literacy of rural minors, who belonged to general and left behind minors’ category, focusing on different age groups. Indeed, the implications of this study might be helpful for the policy makers to look on this issue and think about the future course of action.

Introduction

It is suggested to include the reference of previous studies in this area of research to further expose the research gap. So that the necessity of this study might be further clarified.

Literature review

It is suggested to include the literature review section and relevant hypotheses might be developed.

Materials and Methods

This section was explained adequately.

Results

In Table 4, F-value and Adjusted R2 is at the first and second column. It is suggested to move to the right side of the third column.

Discussion

It is suggested to include the theoretical and practical implications under different sub-headings.

It is suggested to include the limitations and future research also under, separate sub-headings.

Author Response

Response to Reviewer 2 Comments

General Comments

The authors explored an important issue related to the scientific fitness literacy of rural minors, who belonged to general and left behind minors’ category, focusing on different age groups. Indeed, the implications of this study might be helpful for the policy makers to look on this issue and think about the future course of action.

Introduction

It is suggested to include the reference of previous studies in this area of research to further expose the research gap. So that the necessity of this study might be further clarified.

Literature review

It is suggested to include the literature review section and relevant hypotheses might be developed.

Response 1: We are sorry for your suggestion, scientific fitness literacy is the first variable in China's localization, we have analyzed and discussed it as much as possible, but there is not much literature to superimpose for the time being, after all, you can also see, we cite some of our previous research to analyze. However, we've made changes in lines 50-60 and 70-79 to make the introductory part smoother.

Materials and Methods

This section was explained adequately.

Response 2: Thank you for your affirmation.

Results

In Table 4, F-value and Adjusted R2 is at the first and second column. It is suggested to move to the right side of the third column.

Discussion

It is suggested to include the theoretical and practical implications under different sub-headings.

Response 3: Based on your valuable suggestions, we have made corresponding changes to the title of the discussion section and Table 4.

Reviewer 3 Report

The paper is well executed, easy to follow and logically structured. The argumentation and methodology in the paper is convincing. However, a few matters need addressing.

1. The title of your manuscript should be concise. The title of your manuscript is too long, should be concise.

2. E-mail and Affiliations are not complete (city and country names are required) in Line 6-9.

3. The References need be checked.

Lines 346-405. The serial number of the Reference is repeated.

4. Materials and Methods: Well done and referenced. Only one suggestion: Line 141. The Bonferroni test is an important method in this paper, and its implementation steps and principles need to be further explained.

5. Introduction:

What are the possible marginal contributions of this manuscript compared with other studies?

6. Discussion:

The author has made a full discussion in this part. However, it is suggested that the author further divide the Discussion into different sections.

Author Response

Response to Reviewer 3 Comments

The paper is well executed, easy to follow and logically structured. The argumentation and methodology in the paper is convincing. However, a few matters need addressing.

  1. The title of your manuscript should be concise. The title of your manuscript is too long, should be concise.

Response 1: Based on your suggestions, we've made changes to the title.

  1. E-mail and Affiliations are not complete (city and country names are required) in Line 6-9.

Response 2: Sorry, this section has been detailed when we submit it on the website, and it will be automatically generated by the system later. Thanks for the reminder.

  1. The References need be checked.

Lines 346-405. The serial number of the Reference is repeated.

Response 3: Thanks for the reminder that we have removed the extra serial numbers.

  1. Materials and Methods: Well done and referenced. Only one suggestion: Line 141. The Bonferroni test is an important method in this paper, and its implementation steps and principles need to be further explained.

Response 4: Thanks for the reminder, we elaborate on the method on lines 146-148.

  1. Introduction:

What are the possible marginal contributions of this manuscript compared with other studies?

Response 5: We describe this somewhat in the purpose of the study section, as detailed in lines 70-79.

  1. Discussion:

The author has made a full discussion in this part. However, it is suggested that the author further divide the Discussion into different sections.

Response 6: Based on your valuable suggestions, we have amended the headings in the discussion section accordingly

In the language section, we have corrected and polished the grammar of the manuscript. Thank you for your criticism.

Round 2

Reviewer 2 Report

The authors substantially revised the manuscript, based on reviewers' comments. The manuscript may be accepted for publication.